# Improving the Quality of Frozen Fillets of Semi-Dried Gourami Fish (*Trichogaster pectoralis*) by Using Sorbitol and Citric Acid

**DOI:** 10.3390/foods10112763

**Published:** 2021-11-10

**Authors:** Phatthranit Klinmalai, Suwalee Fong-in, Suphat Phongthai, Warinporn Klunklin

**Affiliations:** 1Division of Food Innovation and Bioindustry, College of Maritime Studies and Management, Chiang Mai University, Samut Sakhon 74000, Thailand; phatthranit.k@cmu.ac.th; 2Division of Food Science and Technology, School of Agriculture and Natural Resources, University of Phayao, 19 Phaholyothin Rd, Muang Phayao, Mae Ka 56000, Thailand; suwalee.fo@up.ac.th; 3School of Agro-Industry, Faculty of Agro-Industry, Chiang Mai University, Chiang Mai 50100, Thailand; suphat.phongthai@cmu.ac.th; 4The Cluster of Agro Bio-Circular-Green Industry (Agro BCG), Chiang Mai University, Chiang Mai 50100, Thailand

**Keywords:** gourami fish, sorbitol, citric acid, frozen fish fillets, texture

## Abstract

Semi-dried gourami fish (*Trichogaster pectoralis*) is popularly consumed domestically and exported as a frozen product. This study was conducted to prevent deterioration quality in frozen fish fillets during storage. This research aims to investigate the effects of sorbitol and citric acid at concentrations of 2.5% and 5% (*w*/*w*) of frozen gourami fish fillets compared to the fillets soaked in distilled water on physicochemical properties, such as cooking loss, cooking yield, drip loss, pH, TBARS, color, and texture profile analyses (TPA) during storage at −18 ± 2 °C for a period of 0, 20, and 40 days. The fish soaked in sorbitol and citric acid solutions had significantly (*p* < 0.05) higher protein and fat contents than the control sample. Sorbitol was able to retain moisture in the product; therefore, the drip loss and cooking loss were the lowest, and cooking yield was the highest among other samples (*p* < 0.05). The addition of 5% (*w*/*v*) citric acid in frozen fish fillets can significantly retard the thiobarbituric acid reactive substance (TBARS) (*p* < 0.05) during storage when compared to fish soaked in sorbitol solution kept for the same period. However, the addition of citric acid resulted in low quality in texture and color of frozen fish fillets. The use of sorbitol was the best alternative in frozen fish fillet product due to reducing the negative effects of freezing quality of the products and generating a cryoprotective effect compared to the fillets soaked in distilled water.

## 1. Introduction

Nowadays, Thailand has ample and increasing in exported frozen food or processed food, which is an important industry. Snakeskin gourami fish (*Trichogaster pectoralis*) or Sepat-Siam, also called “leaf fish” because of its shape, is an important economic freshwater fish of Thailand that is normally found in the central region of the country [1]. Gourami fish is normally processed into salted fish or sun-dried fish, consumed widely in Asia and exported as frozen whole gourami fish [1]. The color changing and rancidity of gourami fish fillet normally occurs during chilling storage at 4 °C for seven days. Therefore, freezing is the most common technique used for seafood preservation [2]. However, the freezing process affects product texture, which is related to deformation and disintegration. Frozen foods also need to be thawed prior to cooking or further processing. Thawing aims to restore the quality of fresh products; however, it affects final food quality attributes. Phosphates are then used to improve the functional properties during frozen storage because the phosphates can increase water-holding capacity in fresh products, decrease thawing drip loss, and prevent cooking loss of frozen products [2,3,4,5]. Phosphates also affect sensory attributes, such as juiciness, due to an increase of water-binding properties. However, the excessive use of phosphate as a food additive leads to slimy texture, translucent appearance, and soapy taste in the mouth [3]. Immoderate moisture absorption by using phosphates can lead to consumer fraud. Therefore, phosphates are not allowed for application in most fish products in some countries [2].

The formation of large ice crystals can damage fish protein and lead to quality deteriorations of products during storage, leading to increased drip loss, dehydration, tissue softening, discoloration, and protein denaturation [6,7,8,9]. The quality changes in frozen fish muscle can be reduced by adding suitable cryoprotective substances [10], which can protect muscle tissue from freeze damage. Protein hydrolysate, gelatin hydrolysate, sucrose, and sugar alcohols are used as cryoprotectants for frozen products, which were investigated in frozen surimi [11,12] and frozen fish mince [13].

EFSA and FDA describe sorbitol as a GRAS (generally recognized as safe) humectant, nutritional sweetener, sequestrant, stabilizer, and texturizer in food [14]. Sucrose/sorbitol (1:1) was added in mince fish before frozen and resulted in a greater resistance in surimi gel [11]. The effectiveness of cryoprotection is generally evaluated after storage of frozen foods or multiple freeze/thaw treatments by water-holding capacity, expressible moisture, Ca^2+^-ATPase activity, hydrophobic surfaces, sulfhydryl group and disulfide bond contents, unfrozen water content, and salt-soluble protein concentrations [10,13]. The studies of cryoprotectants mainly focus on surimi products, which have been investigated by many researchers [11,12,15]. Citric acid is inexpensive, non-toxic, and harmless and is widely used in the food industry with a good cryoprotectant effect. According to the Codex GSFA, citric acid may also be used as a food additive in frozen fish and therefore may also be used under the conditions of good manufacturing practices (GMPs) [16]. Citric acid improves physical and sensory characteristics while inhibiting the growth of bacteria in frozen fish products [17]. Therefore, using sorbitol and citric acid is an interesting method for improving the quality of frozen fish fillets.

So far, there are no studies regarding the cryoprotective effect of both sorbitol and citric acid in frozen snakeskin gourami fish fillets. The purpose of this study was to evaluate the cryoprotective effects of sorbitol and citric acid on physicochemical properties of frozen snakeskin gourami fish fillets, such as texture, color, moisture, protein, lipid content, TBARS, pH, drip loss, cooking loss, and cooking yield during storage at −18 ± 2 °C for 0, 20, and 40 days. In addition, the characterizations of frozen/thawed fish fillets were also investigated.

## 2. Materials and Methods

### 2.1. Materials

Semi-dried snakeskin gourami fish (*Trichogaster pectoralis*) was collected from community enterprises in Banpaew, Samut Sakhon province, Thailand. The products were dried in a solar dryer within 24 h (moisture content around 73–74%, pH~6.72). The products were kept in sterile polyethylene bags and immediately moved to the laboratory in a sterile ice-box at the College of Maritime Studies and Management, Chiang Mai University, Samut Sakhon province, within 20-min travel time. Sorbitol and citric acid were purchased from Sigma Aldrich (St. Louis, MI, USA). All chemicals used in the analysis of frozen fillets of dried gourami fish were analytical reagent (AR) grade or the equivalent.

### 2.2. Sample Preparation

Semi-dried snakeskin gourami (*Trichogaster pectoralis*) fillets were cut into 3 × 4 inches. The samples were soaked in two different concentrations comprising 2.5% (*w*/*v*) citric acid, 5% (*w*/*v*) citric acid, 2.5% (*w*/*v*) sorbitol, 5% (*w*/*v*) sorbitol solutions, and distilled water (control) at the weight ratio of 1:1 for 90 min at 4 °C. All the samples were kept in polyethylene bags at −18 ± 2 °C and tested at an interval of every 20 days. Random samples were taken out for analysis to determine quality properties until 40 days. Frozen fish fillets were packed under vacuum and thawed at 4 °C for 24 h with packaging. The sample collection for each analysis was carried out as shown in Figure 1.

### 2.3. Moisture, Protein, and Lipid Content of Frozen Fish Fillets

The moisture, protein, and lipid content of fish fillet samples were analyzed according to AOAC method 935.29, AOAC method 992.15, and AOAC method 930.39 [19], respectively. The samples weighed about 3 g and were placed on the pan before drying in the hot-air oven at 105 °C. After drying, the samples were transferred to the desiccator for cooling and reweighed on the pan with dried samples until sample weight was stable for calculating moisture content. For protein extraction, crude protein content (% total nitrogen × 6.25) was determined by the Kjeldahl method. For lipid analysis, 3–5 g of dried sample was extracted with petroleum ether in a Soxhlet apparatus for approximately 1.5–2 h. Then, the petroleum ether was removed by fractional distillation. The lipid content of samples was determined after drying with lipid to a constant weight in hot air oven at 105 °C.

### 2.4. The pH of Frozen Fish Fillets

The pH of frozen samples was determined according to the standard AOAC method 943.02 [19] and Wangtueai et al. (2021) [18]. It was determined by homogenizing each sample (2 g) in 20 mL of distilled water. The homogenized sample was measured using a pH meter (Ohaus ST3100-B Starter Benchtop, SPCRT Co., Ltd., Parsippany, NJ, USA).

### 2.5. Drip Loss, Cooking Loss, and Cooking Yield of Frozen Fish Fillets

Drip loss was evaluated by modifying the method of Wangteui et al. (2020) [20]. Frozen fish fillets after 20 days of storage were thawed in a refrigerator at 4 °C for 24 h. Excessive water surface was removed using filter paper and weighed before and after thawing. Drip loss was calculated using Equation (1):(1)$$\mathrm{Drip}\;\mathrm{loss}\;\left(\%\right)=\frac{\left(\mathrm{weight}\;\mathrm{before}\;\mathrm{thawing}\;-\;\mathrm{weight}\;\mathrm{after}\;\mathrm{thawing}\right)}{\mathrm{weight}\;\mathrm{before}\;\mathrm{thawing}}\times 100$$

Cooking loss and cooking yield were investigated by weighing thawed fish fillets before and after steaming at 95 ± 2 °C for 15 min until the core temperature reached 70 °C. Cooking loss and cooking yield were determined using the following formulas (Equations (2) and (3)) [21], respectively:(2)$$\mathrm{Cooking}\;\mathrm{loss}\;\left(\%\right)=\frac{\left(\mathrm{weight}\;\mathrm{after}\;\mathrm{soaking}\;-\;\mathrm{weight}\;\mathrm{after}\;\mathrm{steaming}\right)}{\mathrm{weight}\;\mathrm{after}\;\mathrm{soaking}}\times 100$$
(3)$$\mathrm{Cooking}\;\mathrm{yield}\;=\frac{\left(\mathrm{weight}\;\mathrm{after}\;\mathrm{steaming}\;\right)}{\mathrm{weight}\;\mathrm{after}\;\mathrm{soaking}}\times 100$$

### 2.6. Thiobarbituric Acid Reactive Substances (TBARS) of Frozen Fish Fillets

TBARS assay was followed the modification method of Wachirasiri et al. (2017) [22]. Briefly, the internal part of ground fish fillet (0.5 g) was randomly mixed with 2.5 mL of 0.25 N HCl solution containing 0.375% Thiobarbituric acid (TBA) and 15% Trichloroacetic Acid. A random sample was heated in boiling water for 10 min and cooled with tap water. The sample was then centrifuged at 3600× *g* for 20 min. The colored supernatant was determined at 532 nm using a UV-visible spectrophotometer (Biochrom Libra S50 UV/Vis, Cambridge, UK). The TBARS of samples were expressed as mg malondialdehyde per kg of fish fillet compared with a standard curve of malondialdehyde.

### 2.7. Color Measurement of Frozen Fish Fillets

The color values of inner parts of fish fillets were measured using Hunter Lab colorimeter (ColorFlex/EZ, Reston, VA, USA). The values of L*, a*, and b* were used to describe the color characteristics, where L* indicates lightness ranged from 0 (black) to 100 (white); a* represents red with positive a*-value and green with negative a*-value; and b* represents yellow with positive b*-value and blue with negative b*-value. Each frozen/thawed fillet was measured in five different parts. The whiteness index (WI) was calculated using the following Equation (4) [23]:Whiteness = 100 − [(100 − L*)^2^ + a*^2^ + b*^2^]^1/2^(4)

### 2.8. Texture Profile Analysis (TPA) of Frozen Fish Fillets

TPA was performed using a Texture analyzer (TA.XT plus, Stable Micro Systems, Godalming, UK) equipped with a 50-kg load cell by pressing a cylindrical, aluminum probe (diameter, 50 mm; type P/50). Fish fillets were cut into 1-cm cubes, with each side being 3 cm. The muscle fibers oriented horizontally on the fish cubes were analyzed with a speed of 2 mm/min and a trigger point of 0.05 N and compressed with an aluminum cylinder probe until the deformation was reached at 25% [18]. Five independent measurements of samples were taken for each treatment. Textural parameters (hardness, gumminess, springiness, and cohesiveness) were determined from the force-time curve.

### 2.9. Statistical Analysis

A completely randomized design (CRD) was used to analyze the quantitative data. Mean values were compared using analysis of variance (ANOVA) with Duncan’s multiple range test for comparing treatments (*p* < 0.05) in a software package (SPSS ver. 17.0; SPSS Inc., Chicago, IL, USA).

## 3. Results and Discussion

### 3.1. Effect of Sorbitol and Citric Acid on Moisture, Protein, and Lipid Content of Frozen Fish Fillets

The moisture, protein, and lipid content (g/100 g) of the samples stored at different times are shown in Table 1. As storage time prolonged to 40 days, the moisture content of each treatment significantly (*p* < 0.05) decreased, particularly at a low relative humidity level inside the freezer due to evaporation increasing during storage. Moisture content of frozen fish fillets soaked with different solutions was decreased from the first day of storage. For example, the moisture content of frozen fish fillets soaked with distilled water was decreased from 73.20 ± 1.17% (day 0) to 54.98 ± 3.22% (day 40), a decrease of 24.89%, whereas moisture content of frozen fish fillets soaked with 5% sorbitol was slightly decreased by 10.7%. Moreover, moisture content of frozen fish fillets soaked with 5% citric acid was decreased from the lowest initial value compared to other treatments starting at 67.18 ± 0.62% (day 0) and going down to 46.94 ± 1.90% (day 40). The highest moisture losses can be found in frozen fish fillets soaked with citric acid. The small changes in water content of frozen fish fillets soaked with sorbitol might be related to drip loss during the defrosting process; therefore, moisture content after 40 days of storage was slightly changed.

The lipid content of fish fillets soaked with citric acid had the highest value among samples during storage at −18 ± 2 °C for day 0. The protein content of fish fillets soaked with sorbitol and citric acid were higher than a control sample soaked with distilled water at day 0. This is because sorbitol can reduce the migration of water molecules [24,25]. Wachirasiri et al. (2017) [22] and Wangtueai et al. (2021) [20] also reported that water migration to form ice crystals can occur from the dehydration of protein molecules leading to the denaturation of myofibrillar proteins through aggregation and triggered unfolding.

### 3.2. Effect of Sorbitol and Citric Acid on Drip Loss, Cooking Loss, and Cooking Yield of Frozen Fish Fillet

The quality changes in drip loss, cooking loss, and cooking yield of frozen fish fillets during frozen storage at −18 ± 2 °C are presented in Table 2. An increase in drip loss can be associated with decreased water contents (Table 1) during the defrosting process and increased with the storage time due to the formation of ice crystals, resulting in the loss of functionality of cell membranes, which suffer irreversible damage during freezing [26]. Drip loss of frozen fish fillets soaked with 5% sorbitol was the lowest when compared to other samples at 0.68 ± 0.25% (day 20), and at the end of storage, it was quantified at 0.74 ± 0.27% (Table 2). Increasing sorbitol concentration can reduce drip loss in frozen stored fish fillets; in contrast, the drip loss gradually increases with a high citric acid concentration. It can be associated with the moisture content shown in Table 1. Moreover, drip loss can relate to the frozen water and the size of ice crystals in the products. [6,7,9]. Therefore, the decreasing of drip loss after thawing in fish fillets soaked with sorbitol solution can be indicated in the decrease in frozen water and ice crystal size of fish fillets soaked with sorbitol solution.

High cooking loss of frozen products is undesirable and can also be indicative of decreasing in quality owing to exudation of water. Cooking loss of frozen fish fillets soaked with citric acid was remarkably increased during storage (Table 2). On the other hand, sorbitol inhibited the increase of cooking loss, with a strong impact on increasing cooking yield of frozen fish fillets in day 0 and during storage. Sorbitol may prevent the formation of intracellular or intercellular ice crystals [27], which may affect cooking yield

Cooking yield of frozen fish fillets soaked with sorbitol was significantly higher (*p* < 0.05) than the others and increased when increasing the concentration of sorbitol. It may be due to a decrease in the size of ice crystals in frozen fish fillets after being soaked with sorbitol. However, citric acid solution decreases the cooking yield of frozen fish fillets. Moreover, Cho and Song (2017) [28] reported that yields based on weight in retorted frozen seafoods (octopus, squid, and top shell) soaked with sorbitol were increased by 2–5% compared to untreated samples. The incorporation of sorbitol in seafood products can decrease the separation between myotomes and better preserve integrity of the interfibrillar tissue [26]. Recently, Zhang et al. [29] also reported that the sorbitol soaking treatment can inhibit loss from thawing, pressing, and cooking in frozen shrimp.

### 3.3. Effect of Sorbitol and Citric Acid on pH of Frozen Fish Fillet

The pH values of frozen fish fillets soaked with distilled water and sorbitol were neutral, between 6.50 and 6.92 (Table 3). As expected, the pH values of frozen fish fillets with citric acid were 4.36–5.56 because the acidic soaking formed on the surface of fish fillets. The pH value was small but significantly increased (*p* < 0.05) in frozen fish fillets soaked with sorbitol during storage, from 6.50 to 6.92. In contrast, the pH values of frozen fish fillets soaked with citric acid were lower (pH 4.51–5.56) than the other treatments (pH 6.50–6.92). The reduction in pH was probably caused by citric acid, which was used as a soaking solution. The pH of frozen fish fillets soaked with citric acid also increased within 20 days (pH 5.30–5.56) and decreased at day 40 (pH 4.35–4.36) during storage times (Table 3). The pH of samples slightly increased within 20 days because of the increase in volatile bases, such as ammonia, produced by either microbial or muscular enzymes. However, the pH of frozen fish fillets soaked with citric acid decreased due to freezing causes, changes pH values of fish muscle towards a higher acidity, resulting in increasing concentrations of citric acid in the unfrozen water that modified the acid-base equilibrium [21,29]. Moreover, fish fillets were stored at −18 °C for 40 days in a chest freezer, indicating a slow freezing condition. With slow freezing, the unfrozen water is concentrated with protons, which leads to a reduction of pH around structural proteins [30]. Thus, the low pH levels of thawed fish fillets soaked with 2.5% and 5% citric acid influenced drip loss, cooking loss, and cooking yield of fish fillet.

### 3.4. Lipid Oxidation in Frozen Fish Fillet

Thiobarbituric-acid-reactive substances (TBARS) assay is the method to detect lipid oxidation, consisting mainly of malondialdehyde (MDA) as a representative of aldehydes MDA formed through hydroperoxides, which are the initial reaction product of polyunsaturated fatty acids with oxygen [8]. MDA of the sample soaked with distilled water was increased significantly (*p* < 0.05) in the 20 days of storage from 3.76 to 3.93 mg MDA/kg sample and then decreased at day 40 (3.93 to 2.42 mg MDA/kg sample). MDA of the samples soaked with 2.5%, 5% citric acid, and 2.5% sorbitol was not significantly (*p* > 0.05) different before 20 days of storage and decreased at day 40. However, MDA values of frozen fish fillets soaked with 5% sorbitol were high (5.10 mg MDA/kg sample) at day 40 (Figure 2). The increased MDA values in 20 days of storage cause the decomposition of peroxide. In general, several authors indicated a steady increase in MDA over the storage period [31,32,33]. After 20 days of storage, the decline in the TBA level could produce secondary lipid oxidation products (MDA) observed in frozen fish fillets and affect the reaction between MDA and other compounds in the fish fillets [34].

TBARS was low in frozen fish fillet soaked with citric acid and decreased when increasing the citric concentration. Usually, the iron bridges are broken, and iron ions will be oxidation initiators when the pH is low 6.0. However, TBARS of frozen fish fillets increased at pH 4.35. The fluctuations may occur from the interaction of malondialdehyde with other components, such as nucleotides, nucleic acid, proteins, and other aldehydes in fish muscle [35]. Therefore, TBARS of frozen fish treated with 5% citric acid decreased for 40 days. Sorbitol solution was an effective treatment in preventing TBARS values up to 40 days of frozen storage. However, the interaction among amino acids, malondialdehyde, and other non-lipid TBA-reactive substances in samples might affect the TBARS test [33].

### 3.5. Effect of Sorbitol and Citric Acid on Color of Frozen Fish Fillets

Whiteness is a critical parameter for evaluating the quality of fish products. Color values of frozen fish fillets soaked with solutions and distilled water are shown in Table 4. The day 0 of L*, a*, and b* of frozen fish fillets soaked with distilled water were 45.67, −3.39, and 0.96, respectively. After 20 days of frozen storage, the fillets tended to be lighter and more yellowish compared to fresh fillets (*p* < 0.05). Increase in parameters b* and a* represented an increase in yellowness and redness of frozen fish fillets, respectively. The L* and b* values increased significantly after 20 and 40 days of storage, but a* showed slightly changes during storage. During freezing, the yellow color of fish fillets was increased, and red color was increased at the posterior end of the fillets, causing a visible change in appearance (Figure 3). All samples became less red due to the reduced freshness and greyer appearance compared to day 0. An increase in L* value indicated the development of lipid oxidation during storage time, leading to a brighter fillet [36]. Frozen fish fillets soaked with citric acid solution demonstrated effectively increased whiteness (L*) and yellowness (b*) compared to the other samples during storage. The color values of fish fillets soaked with sorbitol solution were similar to the samples soaked with distilled water.

### 3.6. Effect of Sorbitol and Citric Acid Solutions on Texture Properties of Frozen Fish Fillets

Frozen fish fillets were thawed and analyzed the texture properties as shown in Figure 4. The average values of hardness, cohesiveness, springiness, and gumminess displayed significant variations (*p* < 0.05) among different treatments. Hardness of thawed fish fillets soaked with citric acid were increased during storage time. Frozen fish fillets soaked with 2.5% citric acid increased from 97.80 ± 16.68 N (day 0) to 152.28 ± 12.98 N (day 40). Thawed fish fillets soaked with 5% citric acid solution had the highest hardness values (*p* < 0.05) at 226.97 ± 40.16 N. It may be the moisture and greater drip losses of the samples that affected the hardness. Hardness of thawed fish fillets decreased, whereas the liquid loss increased. A low pH was directly related to the loss of texture due to the increase in proton concentration [6,7,9], leading to the weakening of connective tissue and protein denaturation [26]. Citric acid solution increased the hardness and gumminess of restructured fish meat related with the decrease of springiness in thawed fish fillets soaked with citric acid. Frozen fish fillets soaked with citric acid had a low pH (4.35–5.56). The pH values were close to the isoelectric point of fish proteins (pH = 5.5) [37], leading to increase hardness and gumminess. Texture properties of thawed fish fillets soaked with sorbitol were not significantly different compared to the control sample. In addition, there were no significant differences in cohesiveness (*p* > 0.05) among samples.

## 4. Conclusions

The growing consumer demanded for healthy products has encouraged the development of natural cryoprotectant in frozen fish fillets. The addition of citric acid resulted in significantly lower quality in drip loss, cooking yield, and cooking loss and higher quality in hardness and gumminess of frozen fish fillets compared to the control after storage for 40 days due to a low pH in the slow freezing system. Soaking frozen fish fillets in 2.5% (*w*/*v*) and 5% (*w*/*v*) citric acid increased the whiteness of frozen fish fillet due to the solution’s acidity. The sorbitol was the best alternative solution to soak fish fillets before freezing due to reducing the negative effects of frozen products’ qualities, such as cooking yield, drip loss, and pH value. Moreover, texture properties of samples soaked with sorbitol was not significantly different compared to the control sample. In future work, protein denaturation, microstructure, and sensory evaluation of frozen fillets of semi-dried gourami fish (*Trichogaster pectoralis*) should be carried out.

## Figures and Tables

**Figure 1 foods-10-02763-f001:**
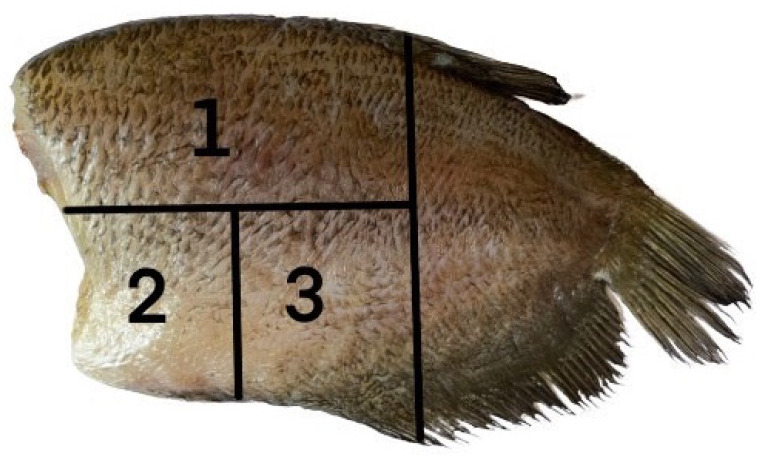
Diagram of the semi-dried Gourami fish used for physicochemical analysis. Each number indicates the part of the fillet used for the different analyses: (**1**) texture and color; (**2**) moisture, protein, lipid content, TBARS, and pH; and (**3**) drip loss, cooking loss, and cooking yield (Modification from Hernández et al. (2009) [18]).

**Figure 2 foods-10-02763-f002:**
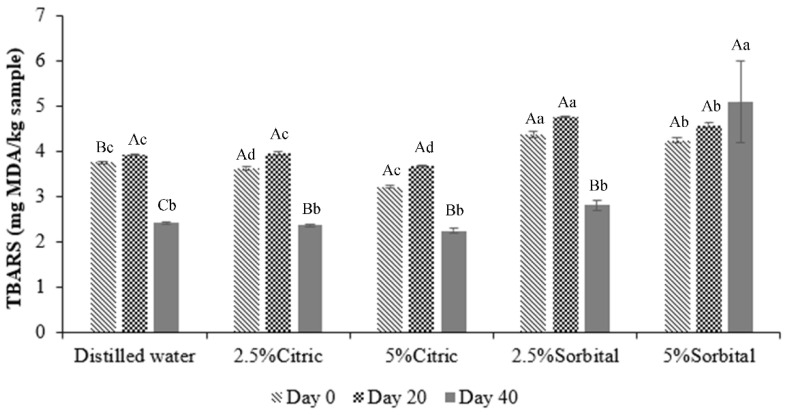
Thiobarbituric-acid-reactive substances (TBARS) of frozen fish fillet soaked with different solutions storage during frozen storage times. Different small superscript letters on the bar are significantly different (*p* < 0.05) with respect to the treatment. Different capital superscript letters on the bar are significantly different (*p* < 0.05) with respect to the period of storage. The results are reported as mean ± SD (*n* = 3).

**Figure 3 foods-10-02763-f003:**
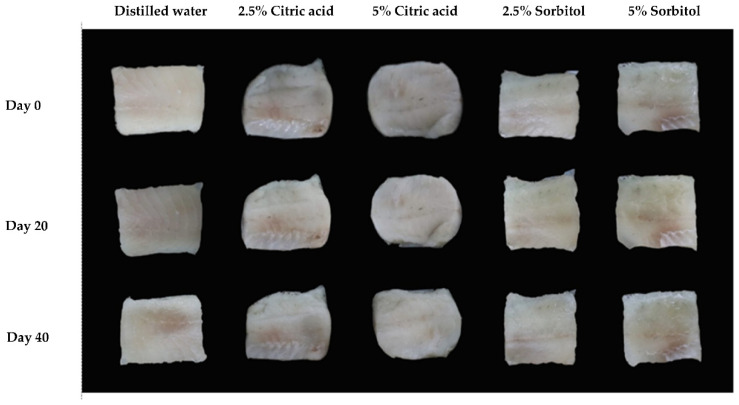
Appearance of frozen fish fillets soaked with different solutions after thawing during frozen storage times.

**Figure 4 foods-10-02763-f004:**
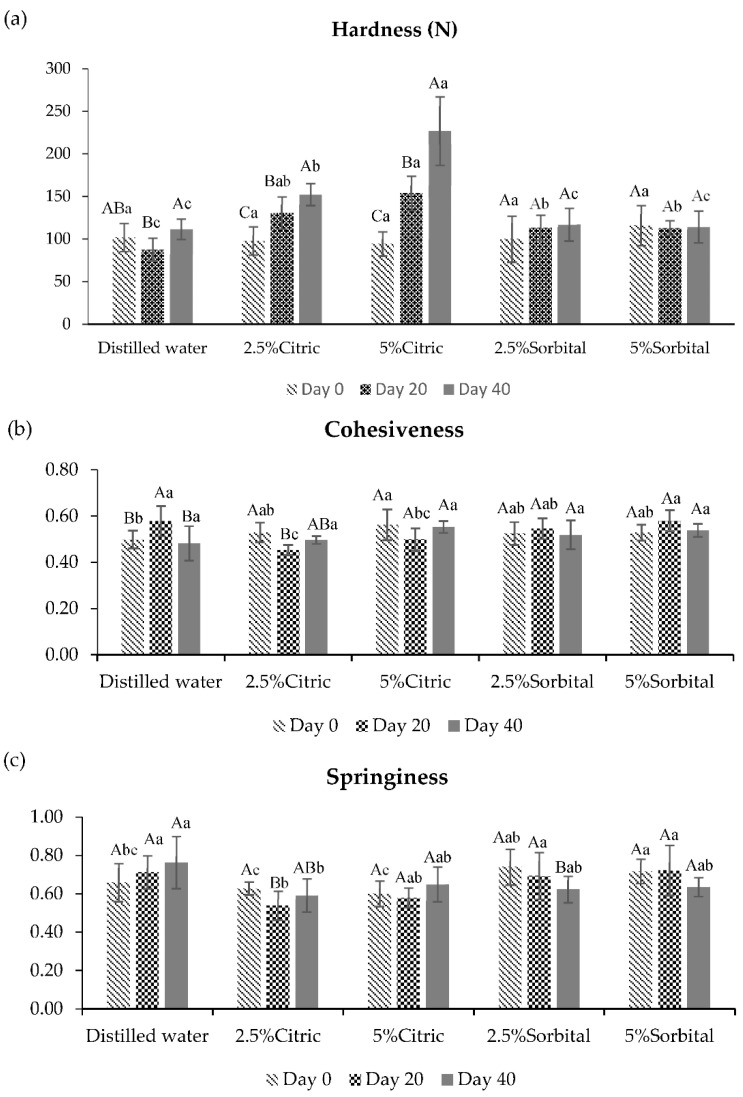
Texture profile analysis (TPA) of frozen fish fillets soaked with different solutions after thawing during storage times. (**a**) Hardness; (**b**) cohesiveness; (**c**) springiness; (**d**) gumminess. Different small superscript letters on the bar are significantly different (*p* < 0.05) with respect to the treatment. Different capital superscript letters on the bar are significantly different (*p* < 0.05) with respect to the period of storage. The results are reported as mean ± SD (*n* = 5).

**Table 1 foods-10-02763-t001:** Lipid, protein, and moisture content of frozen fish fillets soaked with different solutions after thawing during frozen storage times.

Parameter	Treatment	Storage Time		
		Day 0	Day 20	Day 40
Lipid (%)	Distilled water	1.44 ± 0.19 ^Cc^	17.70 ± 1.15 ^Bd^	30.06 ± 1.73 ^Ac^
	2.5% Citric acid	4.89 ± 0.08 ^Ca^	32.97 ± 1.11 ^Ba^	48.44 ± 3.38 ^Aa^
	5% Citric acid	3.62 ± 0.40 ^Cb^	34.74 ± 1.96 ^Ba^	50.00 ± 2.82 ^Aa^
	2.5% Sorbitol	1.88 ± 0.12 ^Cc^	24.42 ± 2.00 ^Bc^	35.80 ± 4.24 ^Abc^
	5% Sorbitol	1.88 ± 0.23 ^Cc^	30.01 ± 1.05 ^Bb^	42.99 ± 6.41 ^Aab^
Protein (%)	Distilled water	12.60 ± 0.09 ^Bb^	15.93 ± 1.53 ^Aa^	17.45 ± 0.90 ^Abc^
	2.5% Citric acid	15.47 ± 1.17 ^Ba^	16.42 ± 0.63 ^Ba^	18.98 ± 0.88 ^Aab^
	5% Citric acid	15.89 ± 0.99 ^Ba^	13.03 ± 0.12 ^Cb^	20.38 ± 0.48 ^Aa^
	2.5% Sorbitol	16.03 ± 0.23 ^Aa^	14.90 ± 1.32 ^Aab^	17.38 ± 2.04 ^Abc^
	5% Sorbitol	15.95 ± 0.38 ^Aa^	16.77 ± 1.17 ^Aa^	16.19 ± 0.95 ^Ac^
Moisture content (%)	Distilled water	73..20 ± 1.17 ^Aa^	61.65 ± 0.52 ^Ad^	54.98 ± 3.22 ^Bb^
	2.5% Citric acid	70.85 ± 0.04 ^Ab^	68.86 ± 0.80 ^Ac^	63.19 ± 2.16 ^Ba^
	5% Citric acid	67.18 ± 0.62 ^Bc^	72.65 ± 1.62 ^Aab^	46.94 ± 1.90 ^Cc^
	2.5% Sorbitol	73.81 ± 1.36 ^Aa^	74.23 ± 1.17 ^Aa^	63.25 ± 2.95 ^Ba^
	5% Sorbitol	73.09 ± 1.24 ^Aa^	71.07 ± 1.17 ^Abc^	66.02 ± 0.94 ^Ba^

Values are mean ± SD from triplicate determinations. Means with different small letters in the same column are significantly different (*p* < 0.05). Means with different capital letters in the same row are significantly different (*p* < 0.05).

**Table 2 foods-10-02763-t002:** Drip loss, cooking loss, and cooking yield of frozen fish fillets soaked with different solutions after thawing during frozen storage times.

Treatment	Storage Time							
	Day 0		Day 20			Day 40		
	Cooking Loss (%)	Cooking Yield (%)	Drip Loss (%)	Cooking Loss (%)	Cooking Yield (%)	Drip Loss (%)	Cooking Loss (%)	Cooking Yield (%)
Distilled water	6.73 ± 2.22 ^Bd^	93.27 ± 2.22 ^Aa^	0.95 ± 0.55 ^Ab^	8.33 ± 0.66 ^Bc^	89.55 ± 2.02 ^Aa^	3.09 ± 1.11 ^Ab^	12.66 ± 2.74 ^Abc^	87.34 ± 2.74 ^Bbc^
2.5% Citric acid	31.64 ± 3.90 ^Ab^	68.36 ± 3.90 ^Bb^	6.12 ± 1.88 ^Aa^	29.23 ± 1.60 ^Ab^	70.77 ± 1.60 ^Bc^	7.16 ± 2.82 ^Aa^	14.38 ± 3.89 ^Bb^	85.62 ± 3.89 ^Ac^
5% Citric acid	38.98 ± 2.89 ^Aa^	61.02 ± 2.89 ^Ac^	6.90 ± 1.47 ^Aa^	31.48 ± 0.97 ^Ba^	68.52 ± 0.97 ^Bd^	8.25 ± 1.06 ^Aa^	19.16 ± 4.94 ^Ca^	80.84 ± 4.94 ^Cd^
2.5% Sorbitol	10.50 ± 1.47 ^Ac^	89.50 ± 1.47 ^Aa^	1.06 ± 0.43 ^Ab^	9.22 ± 2.69 ^Ac^	90.77 ± 2.69 ^Ab^	1.72 ± 0.67 ^Abc^	10.18 ± 3.41 ^Ac^	89.82 ± 3.41 ^Ab^
5% Sorbitol	10.45 ± 2.02 ^Bc^	91.67 ± 0.66 ^Aa^	0.68 ± 0.25 ^Ab^	5.97 ± 2.26 ^Bd^	94.03 ± 2.26 ^Aa^	0.74 ± 0.27 ^Ac^	5.04 ± 1.06 ^Ad^	94.96 ± 1.06 ^Ba^

Values are mean ± SD from triplicate determinations. Means with different small letters in the same column are significantly different (*p* < 0.05). Means with different capital letters in the same row are significantly different (*p* < 0.05).

**Table 3 foods-10-02763-t003:** pH of frozen fish fillets soaked with different solutions after thawing during frozen storage times.

Treatment	Storage Time		
	Day 0	Day 20	Day 40
Distilled water	6.76 ± 0.12 ^Aa^	6.87 ± 0.04 ^Aa^	6.89 ± 0.02 ^Aa^
2.5% Citric acid	4.51 ± 0.07 ^Bd^	5.30 ± 0.07 ^Ad^	4.36 ± 0.06 ^Cc^
5% Citric acid	4.68 ± 0.05 ^Bc^	5.56 ± 0.14 ^Ac^	4.35 ± 0.01 ^Cc^
2.5% Sorbitol	6.53 ± 0.05 ^Cb^	6.92 ± 0.04 ^Aa^	6.66 ± 0.03 ^Bb^
5% Sorbitol	6.50 ± 0.03 ^Bb^	6.69 ± 0.09 ^Ab^	6.67 ± 0.04 ^Ab^

Values are mean ± SD from triplicate determinations. Means with different small letters in the same column are significantly different (*p* < 0.05). Means with different capital letters in the same row are significantly different (*p* < 0.05).

**Table 4 foods-10-02763-t004:** Color measurements of frozen fish fillets soaked with different solutions after thawing during frozen storage times.

Treatment		Storage Time		
		0	20	40
Distilled water	L*	45.67 ± 2.92 ^Bc^	49.06 ± 2.72 ^Ab^	49.59 ± 2.39 ^Ad^
	a*	−3.39 ± 0.39 ^Ca^	−5.28 ± 0.71 ^Aa^	−4.94 ± 0.48 ^Ba^
	b*	0.96 ± 0.43 ^Bc^	3.65 ± 1.58 ^Abc^	4.42 ± 1.18 ^Ab^
	whiteness	45.52 ± 2.82 ^Bb^	48.64 ± 2.63 ^Ab^	49.14 ± 2.39 ^Ad^
2.5% Citric acid	L*	63.91 ± 2.61 ^Bb^	66.67 ± 2.51 ^Aa^	66.51 ± 3.86 ^Ab^
	a*	−3.26 ± 0.59 ^Aa^	−2.66 ± 0.58 ^Bd^	−2.66 ± 0.60 ^Bc^
	b*	4.18 ± 1.52 ^Bb^	8.23 ± 1.28 ^Aa^	7.87 ± 2.19 ^Aa^
	whiteness	63.48 ± 2.50 ^Ba^	65.53 ± 2.42 ^Ab^	65.50 ± 3.87 ^Ab^
5% Citric acid	L*	65.87 ± 3.28 ^Ba^	66.37 ± 2.41 ^Ba^	68.89 ± 2.24 ^Aa^
	a*	−2.54 ± 1.18 ^Bb^	−3.30 ± 0.62 ^Ac^	−2.13 ± 0.75 ^Bd^
	b*	6.58 ± 2.18 ^Aa^	3.05 ± 1.88 ^Bc^	7.59 ± 1.92 ^Aa^
	whiteness	64.6 ± 3.38 ^Ba^	66.05 ± 2.26 ^Ba^	67.84 ± 2.10 ^Aa^
2.5% Sorbitol	L*	45.10 ± 2.97 ^Ccd^	48.23 ± 2.89 ^Bb^	51.51 ± 2.53 ^Ac^
	a*	−3.22 ± 0.81 ^Ca^	−4.16 ± 0.68 ^Bb^	−4.95 ± 0.46 ^Aa^
	b*	1.21 ± 0.87 ^Bc^	4.38 ± 1.84 ^Ab^	3.59 ± 1.55 ^Ab^
	whiteness	50.63 ± 2.99 ^Cbc^	52.82 ± 2.76 ^Bb^	55.19 ± 2.47 ^Ac^
5% Sorbitol	L*	43.62 ± 2.97 ^Cd^	45.38 ± 3.01 ^Bc^	49.22 ± 2.03 ^Ad^
	a*	−2.64 ± 0.96 ^Bb^	−4.40 ± 0.66 ^Ab^	−4.33 ± 0.53 ^Ab^
	b*	1.48 ± 0.90 ^Bc^	3.48 ± 0.80 ^Ac^	3.53 ± 1.40 ^Ab^
	whiteness	43.56 ± 2.86 ^Cc^	45.08 ± 2.91 ^Bc^	48.89 ± 1.96 ^Ad^

Values are mean ± SD from triplicate determinations. Means with different small letters in the same column are significantly different (*p* < 0.05). Means with different capital letters in the same row are significantly different (*p* < 0.05).

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
