# Peer review of "Improving the Quality of Frozen Fillets of Semi-Dried Gourami Fish (Trichogaster pectoralis) by Using Sorbitol and Citric Acid"

_foods, 2021, doi:10.3390/foods10112763_

Round 1

Reviewer 1 Report

This manuscript described the physicochemical characterization of Gourami fish soaked in different chemicals. I have two major concerns. First, more details need to be given on the methodology. For example, (1) the lipid content determination was not even mentioned in the methodology. (2) how to determine protein contents? (3) how to ensure random sampling while measuring different parameters? Specifically, for the measurement of lipid oxidation and color, the surface and inner parts of fish fillet may exhibit very different results. Also, I doubt the accuracy of using a regular pH meter to determine the pH of the fish extracts. Second, the results and discussion could be revised. For example, (1) the cooking yield was not discussed; (2) there is no statistical comparison in TBARS and springiness; (3) the authors should add some discussion related to the regulation of sorbitol or citric acid. Since the conclusion is sorbitol before freezing could improve products quality, is it allowed to be used? If so, what is the allowable limit? In addition, there are some grammar mistakes and typos in this manuscript. Please see my other comments below.

L67: The authors mentioned the samples were dried fish collected locally. However, based on Figure 2, it seems the fish were not dried. Also, based on many manuscripts (see below), the moisture content of dried fish is around 20%; the authors need to explain why their moisture content is all above 60%, which is closed to the undried fish.

Ref 1: Biochemical analysis of Five Dried Fish species of Bangladesh, DOI: 10.3329/ujzru.v31i0.15373.

Ref 2: Biochemical Quality Assessment of Ten Selected Dried Fish Species of North East India, DOI: 10.17148/IARJSET.2016.3107.

Ref 3: Effects of Various Drying Methods on Physicochemical Characteristics and Textural Features

of Yellow Croaker (Larimichthys Polyactis), DOI: 10.3390/foods9020196.

L71: What parameters were analyzed?

L76: Detailed sample preparation protocol is needed. For example, what temperature to soak the samples for 90 min? How to ensure the random analysis?

L78: I guess the authors meant four different solutions. There are only two concentrations: 2.5 and 5%.

L84: How about protein content measurement?

L93: What is the condition of homogenization? After homogenization, the mixture should still have some insoluble particles; how to avoid their interference during pH measurement? Was any centrifugation involved? Also, why did the authors not directly measure the pH of the intact meat using the meat pH probe?

L106-107: What is the difference between Equation 2 and 3? They are the same.

L114: Define random.

L148: It should be a typo here. The moisture, lipid, and protein contents are shown in Table 2.

L150-151: This sentence is conflicted with the data shown in Table 2. For both 5% citric acid and 2.5% sorbitol, the moisture content increased after storing for 20 days. Please explain.

L165-166: Grammar mistake.

L193: How about the discussion about the cooking yield?

L217-218: How is this section related to the hardness?

L219: What is the unsoaked sample?

L260-261: Figure 2 needs to be clearly labeled; otherwise, the readers cannot differentiate each treatment.

L280: Please explain according to a pH value.

L280 and L297: Keep consistency.

L290: What is similar to the pI fish proteins? Change pH = 5.5 to pH 5.5.

In all figure captions, the standard error bar meaning, and the sample size should be indicated. Also, why the springiness in Figure 3 does not contain a significant comparison?

Reviewer 2 Report

Please find in the attached file.

Round 2

Reviewer 2 Report

The authors have addressed the main concerns, but still in table 2 I could not find drip loss for all interval day (day 0, 20, 40) one is missed. 
